# 5-ALA Is a Potent Lactate Dehydrogenase Inhibitor but Not a Substrate: Implications for Cell Glycolysis and New Avenues in 5-ALA-Mediated Anticancer Action

**DOI:** 10.3390/cancers14164003

**Published:** 2022-08-18

**Authors:** Mantas Grigalavicius, Somayeh Ezzatpanah, Athanasios Papakyriakou, Tine Therese Henriksen Raabe, Konstantina Yannakopoulou, Theodossis A. Theodossiou

**Affiliations:** 1Department of Radiation Biology, Institute for Cancer Research, Oslo University Hospital, 0379 Oslo, Norway; 2Institute of Biosciences and Applications, National Center for Scientific Research “Demokritos”, 15341 Aghia Paraskevi, Greece; 3Institute of Nanoscience and Nanotechnology, National Center for Scientific Research “Demokritos”, 15341 Aghia Paraskevi, Greece

**Keywords:** δ-aminolevulinic acid, glycolysis, glycolysis inhibition, metabolism, lactate dehydrogenase, glioblastoma multiforme, glioblastoma treatment

## Abstract

**Simple Summary:**

In the present work, we found that 5-ALA, a natural precursor of heme, can hinder cell glycolysis, which is the main path of energy production for most cancer cells. More specifically, we found that 5-ALA can block an enzyme involved in glycolysis, called lactate dehydrogenase (LDH). We found that 5-ALA has a potency of LDH inhibition comparable to other established LDH inhibitors, such as oxamate or tartronic acid. Nevertheless, 5-ALA has a high accumulation rate in cancers and specifically in the incurable brain cancer glioblastoma multiforme (GBM), which is an important advantage. In fact, because of its high specificity to GBM, 5-ALA is used in the clinic to accurately guide the resection of the tumours, through the light emission of its photoactive product protoporphyrin IX (PpIX). PpIX is the penultimate step in the heme production. Importantly, we show here that continuous administration of 5-ALA killed GBM cells according to their dependence on glycolysis. We additionally found that 20% of externally administered 5-ALA is engaged in the inhibition of LDH, as when LDH was pre-loaded by another inhibitor, tartronic acid, then the cell production of PpIX from 5-ALA was increased by 20%. Since PpIX is an important drug for photodynamic therapy of cancer (excitation by light of PpIX produces oxygen by-products that can kill cancer cells), we additionally discovered that preloading LDH with its inhibitor tartronic acid before performing 5-ALA PDT increases the cancer cell death by 15%.

**Abstract:**

In a course of metabolic experiments, we determined that the addition of δ-aminolevulinic acid (5-ALA) to a panel of glioblastoma multiforme (GBM) cells caused a steep reduction in their glycolytic activity. This reduction was accompanied by a decrease in adenosine triphosphate (ATP) production from glycolysis. These results suggested that 5-ALA is an inhibitor of glycolysis; due to the structural similarity of 5-ALA to the established lactate dehydrogenase (LDH) inhibitors oxamate (OXM) and tartronate (TART), we initially investigated LDH inhibition by 5-ALA in silico. The modelling revealed that 5-ALA could indeed be a competitive inhibitor of LDH but not a substrate. These theoretical findings were corroborated by enzymatic and cell lysate assays in which 5-ALA was found to confer a potent LDH inhibition comparable to that of OXM and TART. We subsequently evaluated the effect of 5-ALA-induced glycolysis inhibition on the viability of GBM cells with diverse metabolic phenotypes. In the Warburg-type cell lines Ln18 and U87, incubation with 5-ALA elicited profound and irreversible cell death (90–98%) at 10 mM after merely 24 h. In T98G, however, which exhibited both high respiratory and glycolytic rates, LD95 was achieved after 72 h of incubation with 20 mM 5-ALA. We additionally examined the production of the 5-ALA photosensitive metadrug protoporphyrin IX (PpIX), with and without prior LDH inhibition by TART. These studies revealed that ~20% of the 5-ALA taken up by the cells was engaged in LDH inhibition. We subsequently performed 5-ALA photodynamic therapy (PDT) on Ln18 GBM cells, again with and without prior LDH inhibition with TART, and found a PDT outcome enhancement of ~15% upon LDH pre-inhibition. We expect our findings to have a profound impact on contemporary oncology, particularly for the treatment of otherwise incurable brain cancers such as GBM, where the specific accumulation of 5-ALA is very high compared to the surrounding normal tissue.

## 1. Introduction

Lactate dehydrogenase (LDH) is a key enzyme that catalyzes the final step in glycolysis [1], reducing pyruvate to lactate and regenerating NAD^+^, which is necessary for continued glycolysis. Since the reverse reaction can also be performed by LDH (e.g., during the Cori cycle stage of gluconeogenesis in the liver [2]), LDH isoforms can be considered key catalysts in NADH/NAD+ redox cycling. The LDH enzyme is a tetramer composed of M subunits encoded by the LDH-A gene or H subunits encoded by the LDH-B gene. Expression of the LDH-A gene is upregulated in many cancers to support the elevated glycolytic activity in these cells, which is known as the Warburg effect [3,4] (Figure 1A). It has been suggested that by switching to a glycolytic phenotype, cancer cells reduce the level of oxidative stress that results from the generation of reactive oxygen species (ROS) along the electron transport chain (ETC) [5]. Moreover, even though the metabolic switch of cancers to glycolysis can be more complicated than mere adaptation to hypoxia [6], the hypoxia-inducible factor 1 (HIF-1), which among others is also induced by hypoxia, instigates RNA encoding of glycolytic enzymes such as aldolase A, phosphoglycerate kinase 1, and pyruvate kinase M [7]. Disruption of glycolysis can thus prove fatal to cancer cells but not to normal cells that are in principle able to compensate with an increase in oxidative phosphorylation-produced ATP [4,8]. This is why LDH has been the target of several medicinal chemistry efforts [9,10,11,12], since it exhibits a higher intrinsic activity to catalyze pyruvate reduction to lactate. Indeed, inhibition of LDH reintroduces oxidative stress, leading to inhibition of tumor progression [13]. Among the best-known linear inhibitors of LDH are oxamic acid (OXM) [14,15] and its analogs [16] and tartronic acid (TART) [14] (Figure 1B). Nevertheless, several other LDH inhibitors, both linear and non-linear, have been reported in the literature [17,18], as overviewed in Rani et al. [19], including FX11 [13,20] and AXKO-0046 [21], the first selective inhibitor of LDH-B. The main problem with the LDH inhibition strategy is selectivity for cancer versus normal cells. There are many normal cells that use aerobic glycolysis, such as those in muscles, the liver, and several functional areas of the brain [22,23]. Inhibition of the *plasmodium falciparum* isoforms of LDH has been additionally employed as an antimalarial strategy [17]

Since LDH is a cytosolic enzyme present in most eukaryotic cells, its release through compromised plasma membranes is routinely used in the lab as an assay of cell death, by necrosis [24]. Apart from the laboratory setting, however, LDH is a useful clinical diagnostic marker [25] in cancer [26] and many other pathologies, such as hemolytic anemia [27], acute myocardial infarction [28], persistent organ failure in acute pancreatitis [29], renal failure [30], and liver failure [31].

δ-aminolevulinic acid (5-ALA; see Figure 1B) is a natural, non-proteinogenic δ-amino acid. In non-photosynthetic eukaryotes, 5-ALA is produced through the condensation of glycine and succinyl-CoA, catalyzed by the mitochondrial 5-ALA synthase [32]. 5-ALA is the first committed step in the biosynthesis of heme, leading to the final precursor of heme, protoporphyrin IX (PpIX), which is a very potent endogenous photosensitizer (PS) used in photodynamic therapy of cancer (PDT) [33,34]. The potential of PpIX as a PS was uncovered by Kennedy and Pottier [35], who used topical administration of high, exogenous 5-ALA concentrations. This overrode the normal feedback inhibition of 5-ALA synthase by heme and consequently overloaded heme biosynthesis, ultimately achieving high PpIX accumulation in tumors. This cancer specificity can be attributed to differences in the porphyrin biosynthetic pathway activity in tumors versus normal tissues [36], including (i) the lack of ferrochelatase (the enzyme responsible for the insertion of Fe into PpIX to convert it into heme) in tumors [37], (ii) the reduced availability of Fe in tumor cells [38] to catalyze the PpIX–heme conversion, and (iii) the increased activity of enzymes [39,40] (e.g., porphobilinogen deaminase) that promote PpIX synthesis in cancer cells.

The specific accumulation of PpIX in cancer cells has found applications outside the obvious in the field of PDT [41]: 5-ALA-derived PpIX photodynamic diagnosis has already been clinically applied to bladder [42], prostate [43], and brain [44] malignancies. In fact, 5-ALA was approved by the U.S. Food and Drug Administration (FDA) in 2017 as an adjunct for the visualization of malignant tissue in grade III and IV glioma (NDA 208630/SN0014); it is currently used in the clinic to precisely guide the resection of brain cancers such as malignant glioma and glioblastoma multiforme (GBM) [45,46]. The accumulation of 5-ALA-derived PpIX in brain lesions in particular is further assisted by the local destruction of the blood–brain barrier (BBB) by neoplasia infiltration that allows for the free influx of 5-ALA, which has otherwise been found to exhibit very low permeability of the intact BBB due to a saturable efflux mechanism at the choroid plexus [47].

It has previously been suggested that 5-ALA enhances aerobic bioenergetics [48], promotes mitochondrial protein expression, and stimulates the expression of heme oxygenase-1 [49] [HO-1], a heme-degrading enzyme. Moreover, 5-ALA in combination with ferrous iron has been shown to reduce hyperglycemia in Zucker diabetic fatty rats [50]. In fact, several clinical studies have revealed a potential role for 5-ALA in glucose control in both prediabetic and diabetic patients with or without [51] the parallel administration of iron [52,53]. In the present study, we examine the efficacy of 5-ALA as a glycolysis inhibitor and the consequences on 5-ALA-based PDT.

## 2. Materials and Methods

### 2.1. Chemicals and Reagents

RPMI 1640 medium without phenol red (Sigma-Aldrich; St. Louis, MO, USA; REF#R7509), Dulbecco’s Phosphate-Buffered Saline with MgCl_2_ and CaCl_2_ (Sigma-Aldrich; St. Louis, MO, USA; REF#D8662), Dulbecco’s Phosphate-Buffered Saline without calcium and magnesium (Sigma-Aldrich; St. Louis, MO, USA; REF#D8537), OptiMem (gibco; REF#11058-022), L-glutamine (Sigma-Aldrich; REF#G7513), Penicillin/Streptomycin (Sigma-Aldrich; St. Louis, MO, USA; REF#P4333), 5-Aminolevulinic Acid or 5-ALA (Sigma-Aldrich; St. Louis, MO, USA; REF#A3785), Tartronic acid (TART, Sigma-Aldrich; St. Louis, MO, USA; REF#86320), Oxamic acid (OXM, Sigma-Aldrich; St. Louis, MO, USA; REF# O3750), Lactate Dehydrogenase (LDH-A, Sigma Aldrich; St. Louis, MO, USA; REF# L1254), β-Nicotinamide adenine dinucleotide (NADH, Sigma Aldrich; St. Louis, MO, USA; REF# N4505), Trizma^®^ base (Sigma Aldrich; St. Louis, MO, USA; REF# T1503), Trizma^®^ hydrochloride (Sigma Aldrich; St. Louis, MO, USA; REF#T3253), Triton™ X-100 solution (Sigma-Aldrich; St. Louis, MO, USA; REF#93443), Sodium pyruvate (Sigma Aldrich; St. Louis, MO, USA; REF# P2256), Thiazolyl Blue Tetrazolium Bromide (MTT, Sigma-Aldrich; St. Louis, MO, USA; REF#M2128), Dimethyl sulfoxide (DMSO, Sigma-Aldrich; St. Louis, MO, USA; REF#276855), Oligomycin A (Oligo, Sigma Aldrich; St. Louis, MO, USA; REF #75351), Carbonyl cyanide 4-(trifluoromethoxy)phenylhydrazone (FCCP, Sigma Aldrich; St. Louis, MO, USA; REF #C2920), Antimycin A (ANTI A, Sigma Aldrich; St. Louis, MO, USA; REF# A8674), Rotenone (ROT, Sigma Aldrich; St. Louis, MO, USA; REF# R8875), Fetal Bovine Serum or FBS (Thermo Fisher Scientific, Oslo, Norway; REF#10270106), 2-Deoxy-D-glucose (2-DG, Sigma Aldrich; St. Louis, MO, USA; REF#D8375), D-glucose (Sigma Aldrich; St. Louis, MO, USA; REF# G8270), L-Glutamine (L-GLUT, Sigma Aldrich; St. Louis, MO, USA; REF# G7513), Seahorse XF Real-Time ATP Rate Assay Kit (Agilent Technologies; Santa Clara, CA, USA; REF# 103592-100), Seahorse XF RPMI Medium (Agilent Technologies, Santa Clara, CA, USA; REF# 103576-100).

### 2.2. Cell Culture

All cells used in the present study were human glioblastoma multiforme (GBM) cell lines, purchased from the American Type Culture Collection (ATCC, Manassas, VA, USA). These cell lines were Ln18 (ATCC^®^, CRL-2610™), U87 (ATCC^®^, HTB-14™), T98G (ATCC^®^, CRL-1690™), and M059K (ATCC^®^, CRL-2365™). All cells were grown in RPMI 1640 without phenol red, supplemented with 10% fetal bovine serum (FBS), 2 mM L-glutamine, and penicillin (100 IU mL^−1^) /streptomycin (100 μg mL^−1^) at 37 °C in a 5% CO_2_ and 95% humidified atmosphere.

### 2.3. Metabolic Assays

In order to study the changes in the metabolism of GBM cells following the in situ addition of 5-ALA, we employed the Seahorse XFe96 analyzer (Agilent Technologies). Briefly, 2–4 × 10^4^ cells were seeded in a seahorse plate and incubated overnight in complete RPMI 1640 medium at 37 °C and a 5% CO_2_ humidified atmosphere. One hour prior to the experiments, the cells had their media changed to XF RPMI at pH 7.4 medium supplemented with 10 mM glucose, 2 mM L-glutamine, and 2 mM sodium pyruvate and were incubated for one hour at 37 °C, 0% CO_2_ atmosphere. The basal oxygen consumption and media acidification rates (OCR and ECAR respectively) of the cell groups were measured in four cycles. After those four cycles, injections of (i) 5-ALA or OXM or TART (500–1000 μM), (ii) OLIGO 1 μM, (iii) FCCP 1 μM, and (iv) ANTIA and ROT 2 μM each were sequentially performed for 3 measurement cycles each.

The ATP contributions of mitochondrial respiration and cell glycolysis in the presence or absence of 5-ALA, TART, and OXM (1000 μM) were deconvoluted and quantified by use of the Seahorse XF Real-Time ATP Rate Assay Kit. In brief, the same protocol as above was followed without the injection of FCCP and with only three basal measurement cycles, while the corresponding final concentrations of OLIGO and ROT/ANTIA were this time 1.5 and 0.5 μM. The ATP rates were determined with use of the Agilent *Seahorse XF Real-Time ATP Rate Assay Report Generator.*

The wells in the bottom row of the seahorse plates were in all cases left devoid of cells and were used as blanks. These were automatically subtracted by the Seahorse analyzer software.

### 2.4. Spectrophotometric Assesment of LDH Kinetics with or without Inhibitors, in Purified Enzyme Assays or Cell Lysates

The reaction mix used was the same for enzymatic solutions and cell lysates. It was a TRIZMA buffer (50 mM) with a final pH of 6.5. The reaction mix also contained 20 mM of sodium pyruvate (the natural substrate for LDH) and 0.4 mg/mL NADH.

*A. enzyme solutions.* A total of 0.5 mL of an aqueous LDH solution was mixed with 0.5 mL reaction mix to start the LDH reduction of pyruvate. The final LDH concentration was always 5 μM. The decay absorbance kinetics of NADH oxidation to NAD+ corresponding to pyruvate conversion to L-lactic acid by LDH were in each case monitored at 340 nm (εNADH 340nm = 0.00622 Lμmol^−1^ cm^−1^), using a Shimadzu UV-2550 UV-VIS spectrophotometer (Shimadzu Corp., Kyoto, Japan). Absorption measurements were recorded every 2 s for 2 min. In experiments with inhibitors, 5-ALA, TART, or OXM were added to and mixed with the aqueous solution of LDH, just before the addition of the reaction mix to initiate the enzymatic reaction. The final concentrations of the inhibitors in the total volume of the reaction mix (1 mL) were 5 mM for 5-ALA and 2.5 mM for OXA or TAR. In the enzymatic assays LDH refers to the LDH-A isoform.

*B. cell lysates.* Cell lysates were obtained from Ln18, T98G, U87, and M059K GBM cells. In brief, cells from confluent T175 flasks were harvested with the use of TrypLE™ Express Enzyme (Gibco, REF# 12604021) and resuspended in ddH_2_O with Triton™ X-100 (0.25%). The cell lysate measurements were performed according to the enzymatic solution measurements; however, in this case, cell lysates equivalent to 10^6^ cells were each time added to the H_2_O half of the reaction mix. The reaction buffer was then added to initiate the enzymatic reactions. The inhibitors (5-ALA, OXA, and TAR) were all added to the same final concentration of 0.75 mM, to the cell lysates prior to the addition of the reaction buffer.

All experiments were repeated independently at least 3 times.

### 2.5. Cytotoxicity by 5-ALA Induced Glycolysis Inhibition

A total of 0.5 × 10^4^ Ln18, U87, and T98G cells were seeded into 96-well microplates and left to grow overnight in a 37 °C, humid 5% CO_2_ environment. Subsequently, the cells were incubated with increasing concentrations of 5-ALA in complete medium (5, 10, and 20 mM) up to 72 h. Standard MTT assays were performed on the appropriate treatment and media control groups at 3 timepoints: 24, 48, and 72 h to assess their cell viability. In brief, the cell media were replaced with 100 µL complete media containing 0.5 mg/mL MTT, and cells were incubated for 1.5–2 h. Next, MTT media were removed from all cell groups and replaced with 100 µL DMSO to dissolve formazan crystals and placed on a shaker for 10 min. The endpoint absorbance was read at 561 nm, using a Tecan spark M10 plate reader (Tecan Group Ltd., Männedorf, Switzerland). Blank values (wells with no cells but with DMSO) were in all cases subtracted. In all experiments, at least 10 parallels were used for each cell group, while the experiments were independently performed 3 times.

All experiments were performed in RPMI media without phenol red and with a normal glucose content of 2 g/L (11 mM), which is closer to a low glucose medium (5.5 mM) than to a high glucose one (30 mM). The seahorse experiments were conducted in the presence of 10 mM D-glucose. We believe that the media we used are optimal for the performance of our experiments, as high glucose media may shift metabolism to glycolysis, while low glucose content may change cell metabolism towards OXPHOS, even leading to cell death in Warburg-type cells. The low glucose media correlate with a fasting blood glucose concentration, while the media we used were closer to postprandial glucose levels in the blood, which is what cells use to cover their bioenergetic requirements.

### 2.6. PDT Experiments

A total of 2 × 10^4^ Ln18 cells (ATCC^®^ CRL-2610™; Manassas, VA, USA) were seeded into 96-well microplates and were incubated overnight in a 37 °C, humid 5% CO_2_ environment. Cells were then pretreated with TAR for 1h (0–6 mM) and then additionally treated with 5-ALA (0–2 mM) for 4 h in OptiMem. Cell irradiation was performed from the 96-well plate undersides by means of a home-built lamp (with four PHILIPS TLD 18W/79 fluorescent tubes) through a red filter. The emission spectrum of the lamp is shown in Appendix A with a peak at ~660 nm and an irradiance of 3 mW/cm^2^. Following irradiation, OptiMem was replaced with complete RPMI 1640medium. Cell viability was assessed by MTT assays 24 h following irradiation, as described in the previous section. The appropriate control groups (both irradiated and dark) were used to determine the cytotoxicity in each case (media controls, TART-only controls, 5-ALA-only controls). In all experiments, at least 8 parallels were used for each cell group, while the PDT experiments were independently performed more than 5 times.

### 2.7. Flow Cytometry

Ln18 cells were seeded in 60-mm petri dishes (8 × 10^5^ cells per dish) and incubated overnight in the normal media at 37 °C and 5% CO_2_ humidified atmosphere. Twenty-four hours after incubation, medium was replaced with 5-ALA (1 mM) in OptiMem for 4 h with or without TART (5 mM). The cells were subsequently harvested with the use of TrypLE™ Express Enzyme, resuspended in PBS (with calcium and magnesium), and kept on ice prior to flow cytometry experiments. A BD LSR II flow cytometer (Becton Dickinson, San Jose, CA, USA) was used. For the detection of PpIX, a 406nm laser was used while the emission was gathered through a bandpass filter at 660/20 nm (channel BV650). Flow cytometry data were analyzed using the software FlowJo v.7.6.1 (Treestar Inc., Ashland, OR, USA).

### 2.8. Computational Methods

The models of LDH-A in complex with pyruvate, OXM, TART, and 5-ALA were based on the high-resolution X-ray crystal structure of LDH-A tetramer in complex with NADH and a synthetic inhibitor (PDB ID: 5W8K) [12]. Missing residue atoms were added using Modeller 9.10 [54], with the sequence of LDH-A isoform 1 from UniProt ID: P00338. The initial bound poses of the ligands were calculated using AutoDock 4.2 [55], after removal of the inhibitors, merging of the non-polar hydrogen atoms, and assignment of Gasteiger charges using AutoDockTools 1.5.4 [56]. The search space was defined by a grid box centered at the active site of each LDH-A monomer and comprised 60 × 60 × 60 grid points of 0.375 Å spacing. For each ligand, 100 docking rounds were carried out with the Lamarckian genetic algorithm and all the default parameters, except for the maximum number of energy evaluations that was increased to 10^6^. The resulting conformations of the ligands were clustered using a 2.0-Å cutoff and for each LDH-A monomer, the highest populated cluster that displayed the lowest-energy pose was selected as the representative ligand-bound conformation.

Atomistic molecular dynamics simulations in explicit solvent were performed using the GPU-accelerated version of PMEMD in AMBER v18 and the ff14SB protein force field [57,58]. For the 4 ligands and the NADH cofactor, GAFF2 parameters with AM1-BCC charges were assigned using the ANTECHAMBER module of AMBER 18 [59]. The protonation state of titratable groups at pH 6.8 were estimated using the H++ server [60], with the preset parameters. In particular, histidine residues 66, 180, 185, and 230 were protonated at N^ε2^ (HΙΕ), His270 was protonated at N^δ1^ (HID), and His192 was at its fully protonated form (HIP) in all monomers of LDH-A. The crystallographic waters were retained, and the initial pose of each ligand was taken from the docking calculations. These tetrameric complexes of LDH-A/NADH with pyruvate, OXM, TART, and 5-ALA were immersed in a truncated octahedral TIP3P water box with a buffer distance of 12 Å around the solute, and then counter ions were added to neutralize the total charge of the system using XLEaP. Initially, energy minimization of each system was carried out with 500 steps of steepest descent and 2000 steps of conjugate gradient without any restraints. Then, positional restraints of 50 Kcal × mol^−1^ × Å^−2^ were applied to all C_α_ atoms, and systems were heated to 300 K within 100 ps of simulation under constant volume (NVT ensemble). The density of the system was then equilibrated under constant pressure of 1 bar and temperature of 300 K through 900 ps of simulation, in which the positional restraints were gradually removed. Two independent production runs of 100 ns were performed in the NPT ensemble with a time step of 2 fs, using different random seed numbers for the assignment of initial velocities at 300 K. The Langevin thermostat with a collision frequency of 2 ps^−1^ was used to regulate the temperature and the Berendsen weak-coupling algorithm with a relaxation time of 1 ps to regulate the pressure. The particle mesh Ewald summation method was used to treat long-range electrostatic interactions with a tolerance of 10^−6^. The real space cut-off was set to 9 Å and all hydrogen atoms were constrained to their equilibrium distance using the SHAKE algorithm. Trajectories were retained every 5000 steps, resulting in 10,000 frames per 10 ps for each simulation and were analyzed using the CPPTRAJ module of AMBER 18 [61]. Figures were rendered using PyMol 2.3.

### 2.9. NMR Experiments

^1^H NMR spectra were recorded on a 500 MHz Bruker Avance spectrometer using a pulse of 9.5 s, pre-acquisition delay 3 s, acquisition time 5 s 36–72 scans to achieve acceptable signal-to-noise ratio and applying presaturation field at the residual water frequency. The solutions were prepared in 50 mM phosphate buffer in H_2_O at pH 6.51. Locking and shimming were performed by means of an external capillary filled with D_2_O.

## 3. Results

In a course of experiments designed to measure the effect of the in situ addition of 5-ALA on the metabolic profiles of four GBM cell lines (U87, T98G, MO59K, and Ln18), we discovered that adding 5-ALA was associated with a dose-dependent drop in the glycolytic activity of all four lines (Figure 1 and Appendix A).

This decline in cellular glycolytic activity led us to hypothesize that 5-ALA could be a de facto inhibitor of glycolysis, especially due to its structural similarity with OXM and TART, which are known inhibitors of LDH (Figure 1); we thus ran metabolic assays employing those inhibitors for comparison. We deduce from the results, summarized in Figure 1, that the greatest drop in glycolytic activity due to 5-ALA was registered in M059K cells (~50% at 1000 μM), followed by Ln18 (~40% at 1000 μM), T98G (~35% at 1000 μM), and finally U87 cells (~10% at 1000 μM). The most efficient inhibitor of glycolysis in the comparative runs presented in Figure 1 was TART, while 5-ALA and OXM showed comparable inhibitory capacities. However, the low aqueous solubility of OXM and its poor cell permeability have to be taken into account.

Even though a marked decrease in extracellular acidification rate (ECAR) was registered as a consequence of administering 5-ALA, OXM, or TART, this was accompanied by a small (5–15%) increase in oxygen consumption rate (OCR) for Ln18, M059K, and U87 cells; no change in OCR was registered for T98G cells (Appendix A).

Following these initial results, we again used the seahorse XFe96 metabolic analyzer to determine changes in ATP production from either respiration or glycolysis from the addition of 1000 μM of 5-ALA or TART. The results are presented in Figure 2; for all the cell lines studied, there was either no change or an increase in ATP production from respiration. In fact, mitochondrial respiration-derived ATP showed a marginally significant increase in U87 cells upon the introduction of 5-ALA (~100 pmol/min) and in M059K cells upon introduction of TART (~75 pmol/min). In all cases, however, the drop in glycolytic ATP production upon introduction of either of the two inhibitors studied was profound. In the case of TART, we recorded reductions of ~−250 pmol/min for U87, ~−375 pmol/min for T98G and M059K, and ~−300 pmol/min for Ln18. The corresponding reductions registered upon introduction of 5-ALA were between 80% and 25% lower: ~−200 pmol/min for U87 and T98G, ~−250 pmol/min for M059K, and ~−140 pmol/min for Ln18.

We also studied the effect of the administration of 5-ALA methyl ester (MetALA) on glycolytic activity in the GBM cell panel and compared it with that of 5-ALA. The results (Appendix A) indicate that in all cases, MetALA conferred a mere ~5% drop in cell glycolysis; in the case of U87, the glycolytic drop was not statistically significant.

### Molecular Modeling of 5-ALA in Complex with LDH-A

With the aim of investigating 5-ALA’s potential activity as an inhibitor of LDH, we employed molecular dynamics simulations (MDs) of a tetrameric model of LDH-A in complex with 5-ALA and NADH. The high-resolution X-ray crystal structure of LDH-A in complex with NADH and an inhibitor (PDB ID: 5W8K) [12] was used for docking 5-ALA in the active site of each LDH-A monomer (Figure 3A–D). Our molecular docking results showed that 5-ALA can be accommodated within the active site of the enzyme, with its carboxylate group interacting with the catalytic Arg168 in the same manner as the pyruvate substrate (Figure 3E), OXM (Figure 3F), and TART (Figure 3G) models. However, the exact binding mode was not similar in all LDH-A monomers, as there was either a salt bridge interaction with Arg168 (chains A and D) or a single hydrogen bond with Arg168 and a second one with His192 (chains B and C). This variability was related to the aminobutanone moiety of 5-ALA, which formed hydrogen-bonding interactions with either Asn137 (chains A, B, and D), Thr247 (chain C), or the NADH cofactor in monomers B, C, and D (Figure 3A–D).

The stability of the predicted bound conformations of 5-ALA in the presence of NADH was investigated using classical MDs in explicit solvent at the 100 ns timescale (see Computational Methods for details). For comparison, similar calculations were performed for LDH-A in complex with NADH and pyruvate, along with two known substrate-like inhibitors, OXM and TART (Figure 1). Their initial conformations were selected according to crystallographic studies of OXM in complex with LDH-A and NADH (PDB ID: 1I10) [62], as shown in the representative close-up views of the LDH-A active site in monomer B (Figure 3G). As a measure of their stability in complex with LDH-A, we used the atomic root-mean-square deviation (RMSD) from the initial docked position. Pyruvate displayed the highest stability as measured by the distribution of the RMSD values in two replicate simulations, followed by OXM (Figure 4). It should be noted, however, that in one molecule of the tetrameric enzyme (chain C), OXM dissociated during the course of the 100 ns MDS. This was also the case for the simulations of TART and 5-ALA, where ligand dissociation was observed in the same monomer of LDH-A (Figure 4). Our results suggest that at least two of the four ligands (chains A and B) bind stably to the active site of LDH-A within the timescale of the simulations. Taken together, our results suggest that 5-ALA can bind at the active site of LDH-A at least as stably as OXM and TART and thus act as an inhibitor of the enzyme.

Following the encouraging results of molecular modelling, we set out to experimentally verify that 5-ALA is indeed an inhibitor of LDH. Initially, we performed enzymatic experiments using pure LDH enzyme solutions (5 μg/mL) in a 50:50 (*v*/*v*) mix of dd H_2_O and TRIZMA buffer (50 mM) with a final pH of 6.5. This pH selection is biologically relevant since it corresponds to the extracellular pH of tumors [63]. The reaction mix also contained 20 mM of sodium pyruvate and 0.4 mg/mL NADH.

On top of the LDH, we added either 5 mM 5-ALA or 2.5 mM TART or OXM. The results of these studies are shown in Figure 5, which shows that all three inhibitors exhibited comparable inhibitions, although 5-ALA was introduced at twice the concentration.

We further determined the initial slopes of all the curves in Figure 5 for their first five data points, corresponding to 8 s overall, to ensure linearity. The slopes (see Table 1) correspond to the *V_max_* of the enzyme at the concentration studied. Upon addition of (i) 5-ALA, the enzyme functioned at ~22% of its *V_max_*, (ii) OXM, it functioned at ~19% of its *V*_max_, and (iii) TART, it functioned at ~14% of its *V_max_*.

In the second phase of the inhibition experiments, we investigated the inhibition of enzymatic activity of LDH from cell lysates. The cells were lysed with an 0.25% aqueous solution of Triton-X 100. Lysates equivalent to 10^6^ cells were added each time to the same reaction buffer for the purified enzyme experiments (see above). In the case of cell lysates, all inhibitors were introduced at a final concentration of 750 μM. The results for all the cell lines studied in this manner are presented in Figure 6. These data make it clear that LDH inhibition varied from one cell line to another; however, TART exhibited the most potent inhibition in all cases. Moreover, it is evident that in the case of lysates, the TART-related curves (NADH absorbance) exhibited an initial rise before starting to decay. Since this initial “kink” was not evident in the enzymatic assays, we attribute it to the spontaneous conversion of intracellular NAD+ to NADH upon addition of the TART-containing solution. Due to this effect, in the cases of OXM and TART, we applied linear regressions to the five points beginning from the maximum value reached before linear decay began to calculate the LDH kinetics slopes. The LDH *V_max_* values for the cell lysate volumes studied are also shown in Table 1. From these data, it can be deduced that U87, Ln18, and M059K contain similar amounts of LDH: approximately 19% more than T98G.

Moreover, from the data in Table 1 we can extrapolate that in U87, LDH operates at 22% of its *V_max_* upon addition of 5-ALA, at ~51% its *V_max_* with the addition of OXM, and at ~17% of its *V_max_* with the addition of TART. The corresponding values for T98G are ~48% for 5-ALA, ~21% for OXM, and ~22% for TART, while for Ln18, they are 32% for 5-ALA, 16% for OXM, and 12% for TART. Finally, LDH in M059K cells functions at 36% *V_max_* with 5-ALA, 22% with OXM, and 20% with TART.

Having established that 5-ALA is a potent inhibitor of LDH, we sought to investigate whether it was also a substrate of LDH; that is, whether LDH modified 5-ALA, as it does in the case of sodium pyruvate.

Our molecular docking results suggest that 5-ALA can bind in a productive, substrate-like conformation within the active site of LDH-A (Figure 3D). In the corresponding conformation, pyruvate is stabilized via an ionic interaction of its carboxylate group with Arg168 so that the carbonyl group is hydrogen bonded with His192 (Figure 3E). During catalysis, a hydride from NADH is transferred to the carbonyl carbon of pyruvate, along with a proton from His192 to the carbonyl oxygen, to produce lactate. A similar conformation of 5-ALA was observed in chain D of LDH-A, although without interacting with all residues of the active site that stabilize pyruvate. The carbonyl group of 5-ALA is in a proper orientation with respect to His192 and NADH for H_2_ addition. However, our molecular dynamics calculations revealed that this bound pose was not stable within the timescale of the simulations, as 5-ALA displayed major conformational rearrangement within the active site; the RMSD from the initial conformation > 4 Å in chain D (Figure 4D). The bound conformations of 5-ALA in the other three protomers did not display a proper orientation and stability for catalysis (Figure 3A–C and Figure 4D). Therefore, our calculations suggest that 5-ALA is not a suitable substrate of LDH-A. Indeed, we performed LDH enzymatic assays similar to those described above but this time substituted sodium pyruvate with 5-ALA. We did not observe any enzymatic activity, and the enzyme kinetics were similar to those of the reaction mixture in the absence of LDH (data not shown).

Nuclear magnetic resonance (NMR) experiments were employed to monitor the molecular integrity of 5-ALA in the presence of LDH/NADH. The spectra were acquired in 50 mM phosphate buffer in H_2_O at pH 6.51 with external D_2_O lock. In the ^1^H NMR spectra (Appendix A), NADH displays a characteristic pair of apparent doublet peaks (*J* = 18 Hz) centered at 2.67 and 2.54 ppm, respectively, which disappear upon conversion of NADH to NAD^+^. On the other hand, pyruvate is recognized by the display of one singlet peak at 2.26 ppm due to the methyl group, while the conversion to lactate can be detected by the emergence of the methyl group signal as a doublet peak at 1.20 ppm (*J* = 7 Hz; Appendix A). The NMR experiments confirmed that the sodium pyruvate/NADH/LDH mixture (20 mM/0.56 mM (or 0.4 mg/mL))/5 μM) results in full consumption of NADH within less than 10 min and concomitant production of lactate, which increases upon the addition of extra NADH (Appendix A). Further, the ^1^H NMR spectrum of an NADH solution (0.56 mM) in the presence of 5-ALA hydrochloride salt (0.75 mM) displays completely unaffected signals for a period of at least 60 min, suggesting that both molecules remain structurally intact at this pH. It must be noted that in unbuffered water, quite visible signs of decomposition of NADH due to the presence of 5-ALA were observed as soon as 10 min after mixing. Consequently, at the suitable pH of 6.51, 5-ALA and NADH in the presence of LDH can be safely examined for 5-ALA conversion into other products with simultaneous NADH spending. This was tested under a range of conditions. Mixing 5-ALA (5 mM/10 μL LDH) with NADH quantities (from 0.45 to 1.57 mM) and monitoring the ^1^H NMR spectrum up to 70 min after addition of NADH revealed no change in 5-ALA peaks, no additional 5-ALA signals, and no conversion of NADH to NAD+ or other products (Appendix A). Therefore, the NMR experiments also confirm that 5-ALA is not substrate for LDH.

Considering these interesting results and intrigued by the compelling and prolific literature on the potential of glycolysis inhibition as an anticancer strategy, we proceeded to investigate the effect of glycolysis inhibition by 5-ALA on the viability of three human GBM cell lines (Ln18, U87, and T98G). The cells were incubated with 5, 10, and 20 mM 5-ALA for three days, and the resulting 5-ALA-induced glycolysis inhibition on their viability was assessed by standard MTT assays at 24, 48, and 72 h following incubation. The results are shown in Figure 7.

The data in Figure 7 show a strong reduction in cell viability at all measurement timepoints for Ln18 and U87 cells. More specifically, both U87 and Ln18 cells show marked reductions of about 90–98% in cell viability for all time points when incubated with 10 mM 5-ALA, which is also the case for incubation at the higher 5-ALA dose of 20 mM. By contrast, T98G cells were not as sensitive to incubation with 10 mM 5-ALA, reaching only LD_40_ after incubation for 72 h. Increasing the 5-ALA concentration to 20 mM, however, did elicit a gradual drop in the viability of T98G, which reached ~30% at 48 h and 5–10% at 72 h. The corresponding metabolic profiles of the three cell lines investigated are presented in Figure 7D. Both U87 and Ln18 appeared to follow a Warburg-type metabolic phenotype with reduced respiration rates, while T98G exhibited high metabolism with respect to both OCR and ECAR. In separate experiments (Appendix A), Ln18 cells were incubated with 5-ALA (5–20 mM) for 24 h, like in the experiments of Figure 7, however 5-ALA was subsequently removed, and the cells were further incubated without 5-ALA for an additional 24 or 48 h. As it can be seen from the data in Appendix A, removing 5-ALA from the cells after 24 h of incubation (in the 10- and 20-mM incubation groups) did not result in abrogation of the cytotoxicity; in fact, the cell survival even decreased further.

Next, we sought to investigate whether LDH inhibition by 5-ALA also meant that less 5-ALA would be available for the production of PpIX downstream of the biosynthetic cycle of heme. To do this, we employed flow cytometry to determine the amount of PpIX generated in Ln18 cells treated only with 1 mM 5-ALA versus cells pretreated with 5 mM TART, which would saturate the intracellular LDH inhibition pockets and ensure that the 5-ALA (1 mM) subsequently administered would be entirely available for conversion to PpIX. The results are shown in Figure 8.

The histograms in Figure 8 make it clear that there was a marked increase (20 ± 2% in the geometrical mean) in PpIX fluorescence when cells were pretreated with TART.

These results prompted us to perform 5-ALA-PDT experiments in Ln18 cells either treated only with 5-ALA (0.5–1.5 mM, 4 h) or additionally pretreated with TART (0.5–5 mM 1 h before 5-ALA and 4 h together with 5-ALA). The results are shown in Figure 9. We showcase a representative outcome in Figure 9A, with 5-ALA treatment (1.5 mM), with or without TART pretreatment (6 mM) and 2.3 J/cm^2^ irradiation (3 mW/cm^2^; for light source spectrum, see Appendix A). These data show that pretreatment with TART enhanced PDT efficacy by ~17(5) %. In Figure 9B, we present the gain in treatment efficiency from all PDT experiments, with significant change in cell death following TART pretreatment. The solid black line represents the median of the data (~16%), while the crossed square symbol represents the mean value (~14%). The box represents one standard deviation from the mean, while the error bars extend to 1.5 standard deviations.

## 4. Discussion

One of the main hallmarks of cancer is the reprogramming of cancer cell metabolism [64], which improves their chances of survival, proliferation, and growth. This reprogramming typically means a shift to a predominantly glycolytic phenotype with a simultaneous suppression of cell respiration, also known as the Warburg effect [3]; however, as Crabtree discovered, it varies across different cancer cells, with some even maintaining high respiration. Even though the benefits of a Warburg type of metabolism are not fully resolved, this type of metabolism has been linked to oncogene activation, such as Ras, Myc, and Akt [65], and a concomitant deficit in onco-suppressor genes, leading to the aberrant production of lactate. Acids like lactate that end up in the extracellular space can decrease the pH of the tumor microenvironment [66], promote tumor invasion [67], and even lead to the suppression of anticancer professional killer cells [68]. Furthermore, as a consequence of the insidious hypoxia in many tumors, particularly as they grow in size, HIF-1 is upregulated, which activates several glycolytic enzyme genes, including LDH, which participates in the redox cycle of NADH/NAD^+^, as it recycles NAD+ from NADH to facilitate the reduction of pyruvate to lactate.

Upon inhibition of LDH, not only are the above advantages suddenly taken away, but an established energy and production machinery also come to an abrupt stop, forcing the cells to resort to respiration for their bioenergetic needs. This sudden switch back to a respiratory metabolism can have serious repercussions for the tumor cells, which in principle have mitochondria adapted to the glycolytic profile. This can lead to severe oxidative stress [13,69], and since in many cases the antioxidant defenses have been decreased as they were not needed in a predominantly glycolytic metabolism, this can lead to cytostasis and even cell death.

This understandably places LDH (one of the key enzymes in glycolysis with a number of known inhibitors) among the most desirable targets for inhibition in the search for an efficient cancer therapy [4,8,70]. In that context, there are numerous inhibitors of LDH, including OXM [14,16] and TART [14], both of which are linear, much like 5-ALA. Apart from these examples, however, there are other potent inhibitors of LDH, some of which are not linear, such as FX11 [13,18]. In the present study, we also tried FX11 alongside 5-ALA, OXM, and TART for comparison. FX11 increased both OCR and ECAR at the concentrations of 5, 10, and 20 μM, while at 50 μM there was a slight reduction in both OCR and ECAR, as revealed by seahorse metabolic assays in M059K and Ln18 GBM cells. In both Ln18 and M059K cells, the drop in OCR at 50 μM FX11 was higher than the corresponding drop in the ECAR, suggesting damage to the cellular bioenergetic machinery. As a result, we decided not to further pursue tests with FX11 in the present study.

In this paper, we demonstrate the ability of 5-ALA to inhibit LDH in a competitive fashion and with a potency comparable to both OXM and TART. The fact that, alongside the decrease in glycolysis rates that 5-ALA conferred to the cell lines investigated, there was a decrease in ATP production by glycolysis that confirms that 5-ALA caused a bioenergetic deficit in these cells, which, with perhaps the exception of U87 cells, was not compensated by OXPHOS-derived ATP.

Both enzymatic assays with human LDH and assays with cell lysates confirm that 5-ALA is an inhibitor of LDH of roughly comparable potency as OXM and TART. In accordance with previous observations [71,72] that 5-ALA is unstable at physiological pH (7.4) in aqueous solutions and degrades quite rapidly, we also observed accelerated rates of conversion of pyruvate to lactate (NADH to NAD^+^) in our enzymatic assays at ~7 pH with 5-ALA present. In parallel, our NMR studies showed a quite rapid degradation of NADH in aqueous solutions with pH values lower than 5.8 [73] and a sustained molecular stability at 6.5 pH for at least 60 min, which was about 30 times longer than the duration of each enzymatic assay. As a consequence, we selected a pH level of 6.5 to ascertain that both 5-ALA and NADH remained stable during the LDH enzymatic assays.

As a direct consequence of the LDH inhibition by 5-ALA, we also discovered, through the 5-ALA conversion to PpIX within the heme cycle, that prior inhibition of Ln18 cells with excess TART increased the amount of intracellular PpIX by approximately 20%. It can hence be deduced that approximately 20% of 5-ALA administered is involved in LDH inhibition and therefore unavailable for conversion to PpIX and then heme. We have also shown that using MetALA instead of 5-ALA reduces the engagement with LDH by 50–80%, depending on the cell line. This result was not unexpected, given that key electrostatic interactions between the carboxylate group of 5-ALA would be disrupted by the introduction of the methyl group, thus lowering the affinity of MetALA for LDH-A.

As delineated in the introduction, 5-ALA is a very valuable prodrug in PDT. Its meta-drug, PpIX, is a very potent PS and, crucially, endogenously generated within the biosynthetic cycle of heme. In contrast to intracellular 5-ALA, which is synthesized in the mitochondria and regulated by a feedback loop linked to the availability of free heme, exogenously supplied 5-ALA bypasses this feedback control and leads to the accumulation of photosensitizing amounts of PpIX. Our PDT results on human GBM Ln18 cells suggest that prior overloading of LDH with excess TART does free up 5-ALA to become available for conversion into PpIX, providing roughly an additional 15% photokilling of Ln18 cells than 5-ALA-PDT alone. This significant observation is in complete agreement with the intracellular production of PpIX with and without TART and suggests that, alongside other modulators that can enhance the 5-ALA PDT [74] outcome, LDH pre-inhibition/overloading can meaningfully contribute to enhanced cancer photokilling.

In a similar study by Golding et al. [75], the use of two glycolysis inhibitors increased the PDT efficacy of 5-ALA on MCF7 human breast carcinoma cells. These two inhibitors were 2-deoxyglucose (2DG) and lodinamine, neither of which is an LDH inhibitor. In fact, they both inactivate mitochondrial hexokinase II, disrupting an early part of the glycolytic machinery, while in addition to glycolysis, lonidamine also profoundly disrupts OXPHOS [76].

Over and above all other implications, however, we believe that the findings presented here can profoundly impact the treatment of GBM, and this is the main reason we modelled the present study on GBM cell cultures. The impact of 5-ALA on the Warburg effect was also studied by Sugiyama et al. [77] on the A549 lung carcinoma cell line. They observed a 5-ALA-induced increase of OXPHOS, which they correlated to an increase in the COX protein expression, and an increase in a suppression of medium acidosis, which they attributed to glycolysis disruption. Furthermore, they showed a significant increase in the generation of superoxide anion radical and the expression of caspase-3, which led to apoptosis. Even though this study produced evidence of disruption of the Warburg effect, there was no mechanistic evidence of the inhibitory action of 5-ALA on cell glycolysis. In addition, LDH inhibition has previously been shown to cause GBM cancer stem cell differentiation and death [78]; however, our findings suggest the possibility of a curative outcome. In the present work, we have clearly shown that incubation of GBM cells with 5-ALA leads to extensive cell death depending on cells’ metabolic profiles. Ln18 and U87 cells that exhibit a Warburg phenotype are almost completely eradicated by a lower dose of 5-ALA (10 mM) as soon as 24 h after incubation. By contrast, T98G, which exhibits a high OCR and ECAR, require double the dose and 72 h of incubation to reach the same level of cell death (LD_90–95_). This is most probably due to the ability of T98G cells to compensate for bioenergetic deficits by increased respiration without detrimental consequences to their survival. Ln18 and U87 cells, however, rely mainly on glycolysis for their energy requirements and hence have adapted their defenses to a low oxidative stress homeostasis, making them more vulnerable to an increase of respiration to compensate for bioenergetic deficits caused by glycolysis inhibition. As outlined in the introduction, 5-ALA is already used in the clinic to precisely guide the resection of GBM tumors through the characteristic fluorescence of its metadrug PpIX, due to its specific accumulation in GBM lesions. This is of course partly attributed to the disparity of key enzyme levels or substrates, such as iron, between normal and cancerous cells, leading to a poor conversion of PpIX to heme in the latter. Most importantly, however, the key reason for the high stockpile of 5-ALA in brain cancer lesions is the fact that an intact BBB (i.e., normal brain tissue) is not permeable to 5-ALA [47]. In contrast, compromised BBB (in and around the GBM lesions) can facilitate the influx and local buildup of 5-ALA. This profoundly higher concentration of 5-ALA in GBM lesions is expected to trigger a profoundly enhanced glycolysis inhibition in comparison to normal tissues and thus lead to selective destruction of the tumor.

In the case of GBM resection guidance, 5-ALA is administered as a single bolus and the fluorescence of the resulting PpIX is used intraoperatively. LDH inhibition from that single 5-ALA bolus is not long enough to elicit a curative effect, due to conversion to PpIX and the fast systemic metabolism of 5-ALA [79]. In our present work, we found that the required dose of 5-ALA to kill GBM cells in culture is between 10 and 20 mM, also depending on the cell metabolic phenotype. In the clinic, the usual dosage given to patients for fluorescence-guided resection of glioblastoma multiforme is 20 mg/kg [45,46]. By example of a person with a mass of 70kg, this corresponds to administration of 1.4g of 5-ALA into a blood volume of ca. 5.5L, i.e., a concentration of 1.5 mM. Assuming a preferable accumulation of 5-ALA in GBM 10-fold than in normal tissue, then this becomes 15 mM, which is comparable to the doses used in our cell studies. This is also the reason we used high concentrations of 5-ALA in the experiments of Figure 7, to mimic the high 5-ALA accumulation in the GBM lesions. It has to be noted that the assumption of a 10-fold increase of 5-ALA concentration in GBM vs. normal tissue is quite modest. In their publication, Stummer et al. [44] showed a ~20-fold increase of PpIX fluorescence in glioma lesions vs. normal tissue (Figure 4 therein, relative peak intensities at 635 nm), while elsewhere, Stepp and colleagues [80] reported the mean PpIX fluorescence intensity in vital glioma tumours to be 100 times higher than in the normal cortex. This increase can be partly attributed to aberrant expression of enzymes participating in the heme biosynthetic cycle but is also due to the lack of the 5-ALA-specific saturable efflux mechanism at the choroid plexus in GBM lesions, a result of the compromised BBB at the tumour sites, as elaborated in the introduction. It is not currently known whether this GBM specificity of 5-ALA is also valid for other LDH inhibitors such as, e.g., OXM and TART, giving 5-ALA a considerable advantage over them for the treatment of GBM by inhibiting LDH, and consequently glycolysis.

As predicted by the molecular modelling and also shown by seahorse measurements on cells pre-treated with 5-ALA (data not shown), 5-ALA is a competitive (i.e., reversible) inhibitor of LDH and when it is removed from the medium the glycolytic rate is quickly restored to its normal levels. This is why only a continuous supply of 5-ALA, at levels that are tolerable by the patient, can result in a sustained inhibition of LDH in the GBM tumour sites, leading the tumours to stasis and/or death and providing a simple solution for the management of this gruesome and currently incurable disease.

## 5. Conclusions

Here, we present, for the first time, evidence that the heme precursor 5-ALA is an inhibitor of the glycolysis enzyme LDH. This is shown by standard LDH enzymic assays on pure enzyme and lysates, as well as with metabolic assays on live GBM cells. The engagement of externally administered 5-ALA with LDH was found to be around 20%, as shown through the increase of PpIX production following the pretreatment with the LDH inhibitor TART. This also has a direct effect on 5-ALA PDT, as again pretreatment of cells with TART increased the 5-ALA PDT efficacy by 15%. Since 5-ALA, and consequently its fluorescence meta-drug PpIX, accumulate selectively in GBM lesions, a sustained infusion of 5-ALA can keep the 5-ALA levels up in the patient’s blood. Furthermore, due to the destruction of the BBB and consequently the saturable efflux mechanism at the choroid plexus, the 5-ALA pool in the patient’s blood can cause increased 5-ALA accumulation in the GBM lesions, resulting in long-term LDH, and hence glycolysis, inhibition, and cancer cell death.

## Data Availability

The data presented in this study are available on request from the corresponding author.

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
