# Peer review of "5-ALA Is a Potent Lactate Dehydrogenase Inhibitor but Not a Substrate: Implications for Cell Glycolysis and New Avenues in 5-ALA-Mediated Anticancer Action"

_cancers, 2022, doi:10.3390/cancers14164003_

Round 1
Reviewer 1 Report
The authors present a study on the effects of 5-ALA as an inhibitor of lactate dehydrogenase in glioblastoma cells. Studying the novel application of well-known photosensitizer 5-ALA is an interesting field. However, several issues must be explained before this paper could be considered for publication in Cancers.
Major points
- Did the authors perform the experiments in a medium containing high or low glucose concentration? Also, did the authors consider that the presence of glucose could affect their findings? It is well-known that glucose in the cell culture medium could change cell metabolism. As was described in several papers, ATP production in vivo relies mainly on OXPHOS. Artificial in vitro conditions, as a consequence of glucose in the medium, favor glycolysis over OXPHOS (Crabtree effect) when the oxygen supply and mitochondrial function are unaffected. I kindly ask the authors to refer to the presence of glucose in the medium at a concentration higher than in physiological conditions and how it may translate into the obtained results.
- Is the 5-ALA irreversible or reversible LDH inhibitor?
- It is unclear why the authors used such a high concentration in the cell viability study (5 mM, 10 mM, and 20 mM), while previous experiments regarding enzyme inhibition were performed in the lower concentration range? By the way, the concentration of 5-ALA is very high; does it possible to obtain this concentration in vivo?
- The authors should better discuss their findings. In literature, several works described the impact of 5-ALA on glycolysis, but none of them was cited in the paper, e.g., 10.3892/or.2013.2945, 10.1038/bjc.2013.391. Please could the authors refer to work presented by Golding et al.; that showed the potentiation of 5-ALA- PDT cytotoxicity against cancer cells when co-treated with glycolysis inhibitors.
- The authors stated that “our findings have a profound impact on contemporary oncology, particularly for the treatment of otherwise incurable brain cancers like GBM, where the specific accumulation of 5 ALA is very high compared to the surrounding normal tissue.” However, it is unclear how their findings can improve cancer patients’ outcomes, especially regarding hard-to-reach places like the brain and highly heterogenic tumors such as GBM.
- Regarding the FACS experiment, it seems like the authors perform one experiment and put the representative data in Figure 8. The correct presentation should include the panel presenting the graph with mean value +/- SD from at least two experiments, the information about the MFI should also be included.
- Figure 9 is unclear. Please could the authors make it more readable?
- What does it mean “RM” in figures 5 and 6? The abbreviations should be explained in the figures’ caption.
Minor points
The nomenclature of parasite species should be in italic (line 85).
Figure 1 could also include the glycolysis pathway scheme.
There are some grammar and typographical errors; please could the authors re-checked the article?
The reference style is incorrect.
Author Response
Reviewer 1
- Did the authors perform the experiments in a medium containing high or low glucose concentration? Also, did the authors consider that the presence of glucose could affect their findings? It is well-known that glucose in the cell culture medium could change cell metabolism. As was described in several papers, ATP production in vivo relies mainly on OXPHOS. Artificial in vitro conditions, as a consequence of glucose in the medium, favor glycolysis over OXPHOS (Crabtree effect) when the oxygen supply and mitochondrial function are unaffected. I kindly ask the authors to refer to the presence of glucose in the medium at a concentration higher than in physiological conditions and how it may translate into the obtained results.
Response: We added the following paragraph in the text (line 234):
All experiments were performed in RPMI media without phenol red and with a normal glucose content of 2 g/L (11mM), which is closer to a low glucose medium (5.5mM) than to a high glucose one (30mM). The seahorse experiments were conducted in the presence of 10 mM D-glucose. We believe that the media we used are optimal for the performance of our experiments, as high glucose media may shift metabolism to glycolysis, while low glucose content may change cell metabolism towards OXPHOS, even leading to cell death in Warburg type cells. The low glucose media correlate with a fasting blood glucose concentration, while the media we used were closer to postprandial glucose levels in the blood, which is what cells use to cover their bioenergetic requirements.
Is the 5-ALA irreversible or reversible LDH inhibitor?
Response: As predicted by the molecular modelling and also shown by seahorse measurements on cells pre-treated with 5-ALA (data not shown) 5-ALA is a competitive, (i.e. reversible) inhibitor of LDH and when it is removed from the medium the glycolytic rate is quickly restored to its normal levels.
The following paragraph was added to the manuscript text (line 676)
“As predicted by the molecular modelling and also shown by seahorse measurements on cells pre-treated with 5-ALA (data not shown) 5-ALA is a competitive, (i.e. reversible) inhibitor of LDH and when it is removed from the medium the glycolytic rate is quickly restored to its normal levels. This is why only a continuous supply of 5-ALA, at levels which are tolerable by the patient can result in a sustained inhibition of LDH in the GBM tumour sites, leading the tumours to stasis and/or death, and providing a simple solution for the management of this gruesome and currently incurable disease”
While the one used previously was deleted (line 608):
It should be noted that 5‑ALA is a competitive inhibitor of LDH; thus, in cells unremittingly incubated with 5‑ALA, the inhibition lasts as long as the incubation. Once the 5‑ALA pool is removed from the cells, the glycolysis quickly returns to its normal levels.
- It is unclear why the authors used such a high concentration in the cell viability study (5 mM, 10 mM, and 20 mM), while previous experiments regarding enzyme inhibition were performed in the lower concentration range? By the way, the concentration of 5-ALA is very high; does it possible to obtain this concentration in vivo?
Response: These concentrations have been used to moderately mimic the high concentrations of 5-ALA in GBM lesions. The concentrations used in our work mimic a mere 10-fold accumulation of 5-ALA in the GBM lesions.
The following paragraph was added in the manuscript text (line 663):
“In the case of GBM resection guidance, 5-ALA is administered as a single bolus and the fluorescence of the resulting PpIX is used intraoperatively. LDH inhibition from that single 5-ALA bolus is not long enough for to elicit a curative effect, due to conversion to PpIX and 5-ALA fast systemic metabolism68. In our present work, we found that the required dose of 5-ALA to kill GBM cells in culture is between 10 and 20 mM, also depending on the cell metabolic phenotype. In the clinic the usual dosage given to patients for fluorescence guided resection of glioblastoma multiforme is 20 mg/kg45,46. By example of a person with a mass of 70kg, this corresponds to administration of 1.4g of 5-ALA into a blood volume of ca. 5.5L, i.e. a concentration of 1.5 mM. Assuming a moderate accumulation of 5-ALA in GBM 10-fold than in normal tissue, then this becomes 15 mM which is comparable to the doses used in our cell studies. This is also the reason we used high concentrations of 5-ALA in the experiments of Fig. 7, to mimic the high 5-ALA accumulation in the GBM lesions.”
- The authors should better discuss their findings. In literature, several works described the impact of 5-ALA on glycolysis, but none of them was cited in the paper, e.g., 10.3892/or.2013.2945, 10.1038/bjc.2013.391. Please could the authors refer to work presented by Golding et al.; that showed the potentiation of 5-ALA- PDT cytotoxicity against cancer cells when co-treated with glycolysis inhibitors.
The following paragraphs have been added:
Line 632:
The impact of 5-ALA on the Warburg effect was also studied by Sugiyama et. al1, on the A549 lung carcinoma cell line. They observed a 5-ALA induced increase of OXPHOS which they correlated to an increase in the COX protein expression, and an increase in a suppression of medium acidosis which they attributed to glycolysis disruption. Furthermore, they showed a significant increase in the generation of superoxide anion radical and the expression of caspase-3 which led to apoptosis. Even though this study produced evidence of disruption of the Warburg effect, there was no mechanistic evidence of the inhibitory action of 5-ALA on cell glycolysis.
Line 624:
In a similar study by Golding et al.75, the use of two glycolysis inhibitors increased the PDT effect of 5-ALA on MCF7 human breast carcinoma cells. These two inhibitors were 2-deoxyglucose (2DG) and lodinamine, none of which is an LDH inhibitor. In fact, they both inactivate mitochondrial hexokinase II, disrupting an early part of the glycolytic machinery, while in addition to glycolysis, lonidamine also profoundly disrupts OXPHOS76.
- The authors stated that “our findings have a profound impact on contemporary oncology, particularly for the treatment of otherwise incurable brain cancers like GBM, where the specific accumulation of 5 ALA is very high compared to the surrounding normal tissue.” However, it is unclear how their findings can improve cancer patients’ outcomes, especially regarding hard-to-reach places like the brain and highly heterogenic tumors such as GBM.
With the new added paragraphs, we hope it is clear how the 5-ALA treatment can substantially benefit the GBM treatments. A sustained infusion of 5-ALA can keep the 5-ALA levels up in the patient’s blood. Furthermore, so due to the destruction of the BBB and consequently the saturable efflux mechanism at the choroid plexus47, the 5-ALA pool in the patients blood can cause increased 5-ALA accumulation in the GBM lesions which can then inhibit LDH in the cancer cells. See also highlighted paragraph at line 656 of discussion.
- Regarding the FACS experiment, it seems like the authors perform one experiment and put the representative data in Figure 8. The correct presentation should include the panel presenting the graph with mean value +/- SD from at least two experiments, the information about the MFI should also be included
Response: Fig. 8 has now been updated according to the suggestions of the reviewer, and the figure legend has been updated accordingly.
- Figure 9 is unclear. Please could the authors make it more readable?
Response: We have now updated Fig. 9 and we have made it more readable with bigger fonts, and larger symbols.
- What does it mean “RM” in figures 5 and 6? The abbreviations should be explained in the figures’ caption.
Response: It has now been clarified in the legend of Figs. 5 and 6 that RM stands for reaction mix.
Minor points:
The nomenclature of parasite species should be in italic (line 85).
Response: The parasite species name has been placed in italic fonts
Figure 1 could also include the glycolysis pathway scheme.
Response: The glycolysis scheme in normal and cancer cells has now been added as part A of scheme 1, while the inhibitor structures have now become 1B.
There are some grammar and typographical errors; please could the authors re-checked the article?
Response: Even though we had the article edited by Scribendi, we have now gone through it once again for grammar and typographical errors.
The reference style is incorrect.
Response: We have now corrected the references using the Mdpi endnote format.
Reviewer 2 Report
In their manuscript, Grigalavicius, et al. reports that δ-aminolevulinic (5-ALA) inhibits lactate dehydrogenase activity (LDH) and therefore hinders glycolysis and viability of brain cancer glioblastoma multiforme (GBM). The authors showed that 5-ALA administration in GBM cells specifically inhibited glycolysis and ATP generated from glycolysis. They further performed molecular modeling of 5-ALA binding pattern on LDH and experimentally proved that 5-ALA inhibits LDH activity. The paper is well-structured. A few experiments need to be done to improve the quality of this paper.
1. In addition to seahorse experiment, additional measurements are needed to show that glycolysis is inhibited. For example, what are the changes of glucose consumption and lactate production upon the administration of 5-ALA? To be more accurate, 13C labeled pyruvate could be used to trace the conversion between pyruvate and lactate and determine whether the activity of LDH is altered in the context of live cells.
2. The authors showed that 5-ALA increased the cell viability of GBM cell line. Could the viability be rescued by restore LDH activity?
Author Response
Reviewer 2
In their manuscript, Grigalavicius, et al. reports that δ-aminolevulinic (5-ALA) inhibits lactate dehydrogenase activity (LDH) and therefore hinders glycolysis and viability of brain cancer glioblastoma multiforme (GBM). The authors showed that 5-ALA administration in GBM cells specifically inhibited glycolysis and ATP generated from glycolysis. They further performed molecular modelling of 5-ALA binding pattern on LDH and experimentally proved that 5-ALA inhibits LDH activity. The paper is well-structured. A few experiments need to be done to improve the quality of this paper.
- In addition to seahorse experiment, additional measurements are needed to show that glycolysis is inhibited. For example, what are the changes of glucose consumption and lactate production upon the administration of 5-ALA? To be more accurate, 13C labeled pyruvate could be used to trace the conversion between pyruvate and lactate and determine whether the activity of LDH is altered in the context of live cells.
Response: We thank the reviewer for their suggestion. Indeed, 13C labelled pyruvate is another method of showing, what we believe we have shown unambiguously and in three different ways. The most indefatigable evidence is this from the experiments with the purified LDH enzyme, where there is nothing more in the mix apart from the enzyme the substrate, the inhibitors and the NAD+/NADH pair. This is the standard assay for measuring LDH. We also replicated these experiments with cell lysates from 3 different cell lines instead of purified enzyme, but also in intact cells using the seahorse. The decrease of the pH in the seahorse experiments shows the reduction in lactate in the presence of 5-ALA, which is also dose-dependent. We believe that these 3 experiments we chose confirm our hypothesis without the shadow of a doubt and the performance of the 13C pyruvate experiments would only prove what was already proven, and hence from our point of view are redundant.
- The authors showed that 5-ALA increased the cell viability of GBM cell line. Could the viability be rescued by restore LDH activity?
Response: The short answer is no. The viability could not be restored. We conducted experiments to that end on LN18 cells where the 5-ALA was removed after 24h incubation. Since 5-ALA is a competitive, reversible inhibitor of LDH (see also point 2 of reviewer 1), the LDH activity was restored within minutes of the 5-ALA experiments. In the following 24 and 48 h of incubation without 5-ALA, not only there was no abrogation of the cytotoxicity in the 10- and 20-mM groups but there was a further decrease of cell survival. The results of these experiments have been presented as Fig. S6, while the following paragraph has been added to the text in line 506, highlighted yellow:
“In separate experiments (Fig. S6), LN18 cells were incubated with 5-ALA (5-20 mM) for 24h like in the experiments of Fig. 7, however 5-ALA was subsequently removed and the cells were further incubated without 5-ALA for further 24 and 48h. As it can be seen from the data in Fig. S6, removing 5-ALA from the cells after 24h of incubation (in the 10- and 20-mM incubation groups) did not result in abrogation of the cytotoxicity; in fact, the cell survival even decreased further.”
Round 2
Reviewer 1 Report
This article's quality has improved, and the authors responded to most of my concerns.
However, still, there are several questions needed to be resolved. I do not know if the authors understand my question regarding using LDH inhibition in GBM therapy. In this question, I did not mean to explain the possible use of 5-ALA in treating GBM, but what is the idea of inhibiting the LDH enzyme? Could the authors answer how LDH inhibition in vivo might contribute to GBM therapy? Can the inhibition of one enzyme have a "profound impact on contemporary oncology"? It is unclear how inhibition of LDH by 5-ALA might improve clinical outcomes. The different studies showed that glycolysis inhibitor enhanced 5-ALA-mediated PDT activity. Thus, what is more, important in treating GBM, inhibiting LDH enzyme by 5-ALA, and decreasing the concentration of 5-ALA available for conversion to PIXP or PDT with 5-ALA? Besides, a high concentration of 5-ALA to inhibit the enzyme is needed. I understand the explanation that 5-ALA might accumulate selectively and at higher concentrations in tumor tissue. However, how do the authors know that the concentration of 5-ALA is 10 times higher in neoplastic tissue while this statement has no reference? What about the saturable efflux mechanism present at the choroid plexus? I highly appreciate the discovery that 5-ALA can inhibit the LDH enzyme. However, should this result not be understood as describing another property of 5-ALA that may serve as OXM and TART as an enzyme inhibitor and not as a drug that, by inhibiting LDH, may revolutionize the treatment of GBM?
Author Response
Point by point answer to Reviewer 1’s concerns
1. However, still, there are several questions needed to be resolved. I do not know if the authors understand my question regarding using LDH inhibition in GBM therapy. In this question, I did not mean to explain the possible use of 5-ALA in treating GBM, but what is the idea of inhibiting the LDH enzyme? Could the authors answer how LDH inhibition in vivo might contribute to GBM therapy? Can the inhibition of one enzyme have a "profound impact on contemporary oncology"? It is unclear how inhibition of LDH by 5-ALA might improve clinical outcomes.
Response: As explained in the text, inhibition of LDH is expected to disrupt glycolysis in the GBM tumours, leading them to death. This is going to work due to the selective accumulation of 5-ALA in the GBM lesions, due to the compromise of the BBB in the lesions and consequently local destruction of the 5-ALA specific choroid plexus efflux pump, which keeps 5-ALA out of the normal brain tissue. As it appears from clinical publications on 5-ALA diagnosis which we have now cited, the selective accumulation of 5-ALA in the malignant glioma lesions could be as high as 100:1 compared to normal tissue.
2. The different studies showed that glycolysis inhibitor enhanced 5-ALA-mediated PDT activity. Thus, what is more, important in treating GBM, inhibiting LDH enzyme by 5-ALA, and decreasing the concentration of 5-ALA available for conversion to PIXP or PDT with 5-ALA?
Response: Our suggestion is not to use 5-ALA to inhibit LDH and then perform PDT (although that could also be possible after a prolonged inhibition of LDH with 5-ALA killing off many cells by glycolysis inhibition and then killing the rest by PDT).In fact here we propose two different treatments: i) Long 5-ALA incubation without PDT application for inhibiting glycolysis in Warburg GBM cells and ii) Prior inhibition of LDH with another inhibitor e.g. tartronic acid, and then incubation with 5-ALA so more 5-ALA is going to be available for conversion to PpIX for the needs of PDT
3. Besides, a high concentration of 5-ALA to inhibit the enzyme is needed. I understand the explanation that 5-ALA might accumulate selectively and at higher concentrations in tumor tissue. However, how do the authors know that the concentration of 5-ALA is 10 times higher in neoplastic tissue while this statement has no reference?
Response We have now cited references and in fact it can be much higher than 10 times. What about the saturable efflux mechanism present at the choroid plexus? This(the efflux mechanism) is destroyed in the vicinity of the GBM lesions so the lesions are flooded with 5-ALA.
4. I highly appreciate the discovery that 5-ALA can inhibit the LDH enzyme. However, should this result not be understood as describing another property of 5-ALA that may serve as OXM and TART as an enzyme inhibitor and not as a drug that, by inhibiting LDH, may revolutionize the treatment of GBM?
Response: The fact that there is a specific efflux mechanism for the efflux of 5-ALA in the normal brain is an advantage for the use of 5-ALA in GBM therapy. As mentioned above we know that in compromised BBB the GBM lesions are flooded with 5-ALA as opposed to the normal brain so extended administration of 5-ALA will only kill the GBM tissue. We do not know how this works for OXM or TART. We do not know their ratio of accumulation in diseased vs. normal tissue, and we do not if there are any mechanisms keeping them out of the intact BBB.
We have now added the following paragraph to the text in line 680:
”It has to be noted that the assumption of a 10-fold increase of 5-ALA concentration in GBM vs. normal tissue is quite modest. In their publication Stummer et al.[1], showed a ~20-fold increase of PpIX fluorescence in glioma lesions vs normal tissue (Fig. 4 therein, relative peak intensities at 635 nm) , while elsewhere Stepp and colleagues[2], reported the mean PpIX fluorescence intensity in vital glioma tumours to be 100 times higher than in the normal cortex. This increase can be partly attributed to aberrant expression of enzymes participating in the heme biosynthetic cycle, but is also due to the lack of the 5-ALA-specific saturable efflux mechanism at the choroid plexus in GBM lesions, a result of the compromised BBB at the tumour sites, as elaborated in the introduction. It is not currently known whether this GBM specificity of 5-ALA is also valid for other LDH inhibitors such as e.g. OXM and TART, giving 5-ALA a considerable advantage over them for the treatment of GBM by inhibiting LDH, and consequently glycolysis.”
References:
- Stummer, W.; Stocker, S.; Wagner, S.; Stepp, H.; Fritsch, C.; Goetz, C.; Goetz, A.E.; Kiefmann, R.; Reulen, H.J. Intraoperative detection of malignant gliomas by 5-aminolevulinic acid-induced porphyrin fluorescence. Neurosurgery 1998, 42, 518-525; discussion 525-516, doi:10.1097/00006123-199803000-00017.
- Stepp, H.; Beck, T.; Pongratz, T.; Meinel, T.; Kreth, F.W.; Tonn, J.; Stummer, W. ALA and malignant glioma: fluorescence-guided resection and photodynamic treatment. J Environ Pathol Toxicol Oncol 2007, 26, 157-164, doi:10.1615/jenvironpatholtoxicoloncol.v26.i2.110.